# Non-typhoidal *Salmonella* contributes to gastrointestinal infections in Morogoro: Evidence from patients attending Morogoro regional referral hospital in Tanzania

**Anacleta Cuco[1], Ernatus Mkupasi[1], Alexanda Mzula[2]*, Robinson Mdegela[1]**

**1** Department of Veterinary Medicine and Public Health, College of Veterinary Medicine and Biomedical Sciences, Sokoine University of Agriculture, Morogoro, Tanzania, **2** Department of Microbiology, Parasitology and Biotechnology, College of Veterinary Medicine and Biomedical Sciences, Sokoine University of Agriculture, Morogoro, Tanzania

\* amzula@sua.ac.tz

## Abstract

### Introduction

*Salmonella* is one of the most common causes of food-borne outbreaks and infection worldwide. Non-typhoidal *Salmonella* (NTS) infections are common and remain a significant public health problem among important bacterial foodborne diseases. The current study aimed to establish the Non typhoidal *Salmonella* infection and antimicrobial resistance status among selected patients at Morogoro Regional Referral Hospital (MRRH), Morogoro Region, Tanzania, to inform clinical care management and public health interventions.

### Methodology

A cross-sectional study was conducted using medical records and samples were collected from hospitalised and outpatients between October and December 2021. A total of 153 participants were enrolled in the study and 132 consented to being sampled. The collected samples were analysed using standard microbiological techniques. The isolates were subjected to molecular genotyping, where Polymerase Chain Reaction (PCR) was performed targeting the 16S rDNA gene. PCR products were then submitted for sequencing to establish phylogenetic relatedness. Antimicrobial susceptibility testing and resistance genes screening were also conducted.

### Results

The phylogenetic analysis identified two *Salmonella* serovars; *Salmonella* Enteritidis and *Salmonella* Typhimurium. The isolates were from four adults and seven children patients. The isolates were tested against six antimicrobial agents: tetracycline, trimethoprim, gentamycin, ciprofloxacin, ampicillin and cefotaxime. Further antimicrobial assays were performed by screening 10 antimicrobial resistance genes using PCR. Overall, the highest resistance was observed in ampicillin (100%), whereas the lowest resistance was recorded for ciprofloxacin and gentamicin (9.1%). In addition, four (36.4%) of the isolates were

**Data Availability Statement:** All data generated or analysed during this study are included in this article.

**Funding:** This work was supported by the Inter-University Council for East Africa (IUCEA) (Grant no. IUCEA/ELP/004 to AC) and UNESCO-TWAS Fellowship for Research and Advanced Training (Grant no.3240322679 to AC). The funders had no role in study design, data collection and analysis, decision to publish, or preparation of the manuscript.

**Competing interests:** The authors have declared that no competing interests exist.

resistant to cefotaxime and three (27.3%) to tetracycline and trimethoprim. The isolates also exhibit the presence of resistance genes for *sulfamethoxazole 1&2*, *tetracycline (tet) A&B*, *Beta-lactamase $_{CTXM}$*, *Beta-lactamase $_{TEM}$*, *Beta-lactamase $_{SHV}$*, *Gentamycine*, *Acra and acc3-1* in different occurrences. The overall prevalence of *Salmonella* species in Morogoro region was 8.3% (11/132) with *Salmonella* Enteritidis *and Salmonella* Typhimurium being the only serovars detected from adults and children stool samples.

## Conclusion

Our investigation showed that both children and adults had been exposed to *Salmonella* spp. However, the occurrence of NTS was higher in children (5.3% (7/132) compared to adults (3.0% (4/132). To stop zoonotic infections and the development of antimicrobial resistance in the community, this calls for Infection Prevention and Control (IPC) and stewardship programmes on rational use of antimicrobials in both health facilities and at the community level.

## Author summary

Typhoid fever is claimed to prevail in Morogoro region. Most patients with stomach pain and or diarrhoea go for a diagnosis of typhoid despite the existence of other etiological agents causing the same clinical signs. Sample culture and isolation of bacteria have been a gold-standard method for the detection of disease-causing agents. However, this approach is rarely made by most hospitals, with the exception of regional and district hospitals in the public sector and big hospitals in the private sector. They always rely on the widal test, which suffers from significant limitations in its sensitivity and specificity. It is, therefore, difficult to identify appropriate etiological agents and assess their susceptibility to antimicrobials. Because of that, the contribution of non-typhoid salmonellosis is not featured. The authors have assessed the occurrence of non-typhoidal *Salmonella* in hospitalized and outpatients of all age groups at Morogoro Regional Hospital as a modal to establish its contribution to gastrointestinal infections in the region through both conventional and molecular approaches. It was observed that the infection rate was high in children, and the non-typhoidal *Salmonella* serovars had considerable resistance to the commonly used antimicrobial. This calls for further surveillance of these neglected non-typhoid salmonella in clinical settings

## 1. Introduction

*Salmonella* infection is a major public health challenge of worldwide concern in both industrialised and developing countries and has contributed to an increased economic burden on health systems [1]. Non-typhoidal *Salmonella* is a common disease in developing countries, particularly in sub-Saharan Africa, due to problems of poor sanitation, low hygiene, and lack of access to safe water and food, contributing to around 9% of deaths in children under 5 years of age [2,3]. Most of the *Salmonella* spp. infected individuals present diarrhoea, high fever with prostration, and stomach cramps. Symptoms typically appear 6 hours to 6 days after exposure and last 4 to 7 days [1]. Antimicrobials are only used to treat people with severe illnesses with life-threatening symptoms [4]. In severe cases, hospitalisation may be required. Every year,

approximately 600 million people, or nearly one-tenth of the global population, become ill, and 420 000 die, resulting in the loss of 33 million disability-adjusted life years (DALYs), accounting for 22.2% of DALYs due to diarrhoea [5].

According to the White-Kauffmann-Le Minor scheme [6], *Salmonella* has over 2600 serotypes, with the majority (approximately 1600 serotypes) belonging to the subspecies enterica. Though many serotypes (over 200) have been identified as pathogenic agents in humans, the two most common serotypes causing human salmonellosis are *Salmonella* Enteritidis (*S.* Enteritidis) and *Salmonella* Typhimurium (*S.* Typhimurium) [7]. Both Serotypes (*S.* Typhimurium and *S.* Enteritidis) produce the symptoms of gastroenteritis and bloody diarrhoea [8]. Therefore, many strain types make it challenging to diagnose the condition rapidly and accurately [9]. In Tanzania, there are few reports of the occurrence of *Salmonella* spp. infection in both humans and animals [10–12] and *Salmonella* infection in Tanzania, concretely in Morogoro, may be higher than expected. Healthcare services in low- and middle-income countries are facing significant challenges due to antimicrobial multi-drug resistance, which is increasing rapidly due to increased global human mobility and irrational use of antimicrobials in the health and livestock sectors [13]. Human infections with multi-resistant bacterial strains, including *Salmonella*, can result in prolonged hospitalisation and increased mortality, especially in children with invasive infections [14]. Diarrhoeic cases in patients admitted to Morogoro referral hospital are common, but the aetiologies were not well elucidated. The contribution of NTS was not well known; hence, it often goes unnoticed despite its disastrous effects in terms of human suffering and economic costs. Moreover, before this study, limited evidence existed on the incidence of NTS gastroenteritis in MRRH, Tanzania. Therefore, this study aimed to establish the status of NTS and its antimicrobial profile at MRRH, Tanzania, to inform its existence and treatment consideration.

Estimating the occurrence of NTS among diarrhoeic hospitalised and outpatients in MRRH is paramount for planning strategies and developing preventive and control strategies to reduce admissions caused by diarrhoeic cases. In addition, the antimicrobial susceptibility profiles of the isolated NTS assessed will guide updating treatment guidelines in public health interventions.

## 2. Materials and methods

### 2.1 Ethics statement

In the present study, ethical clearance was granted by the National Institute of Medical Research, Dar es Salaam, Tanzania (Ref No.: NIMR/*HP/R.8a/Vol.IX/3808*-14[th] October 2021). Permission to carry out the study was also sought from the regional health management team (RHMT) (DC.65/245/01 - 18[th] September 2021), the Morogoro region district office and the management of the MRRH. Ethical approval for the study was given by the ethical committees of Sokoine University of Agriculture, Tanzania, with reference No SUA/AMD/R.1/8/763, approved on 18th October 2021. All study participants were informed about the study objectives and the procedures involved in enrollment participation. Before collecting faecal specimens, signed informed consent was obtained from each adult individual. Additionally, the parent or legal guardian signed a written consent form for minors under 18 years of age.

### 2.2 Description of the study area

The study was conducted in MRRH from October 2021 to December 2021. Morogoro town is the headquarters of Morogoro region, with a population of 986,678 in 2022 [15], located in the eastern part of Tanzania, 196 kilometres (122 mi) west of Dar es Salaam. Morogoro town lies

at the base of the Uluguru Mountains [16]. Morogoro residents receive health services from health institutions owned by the government and private sectors.

The study area was chosen based on the more significant number of patients attending the facility in the region. Hence, patients attending the MRRH presenting with signs suggestive of *Salmonella* spp. infectious, such as stomach pain symptoms and gastroenteritis, were recruited. The participants were characterised by their age, gender, marital status and occupation. The 132 stool samples from patients with gastrointestinal problems without diarrhoea as the first symptom admitted at the hospital and outpatients who went to the hospital complaining about diarrhoea originating in Morogoro were analysed. In this study, age groups were classified as follows; children under five years old, children (5–8 years), teenagers (13–19 years), younger (20–21 years), adults (22-60years) and elders (above 60 years).

### 2.3 Sampling and patient recruitment

Purposive sampling based on clinical signs and symptoms was used to recruit participants in this study. Participants from the four hospital wards, recovery wards (male and female), surgical wards (male and female), maternity ward (females only), laboratory and outpatients were selected for recruiting patients with assistance from the clinician depending on clinical presentation and their willingness to participate in the study. Study participants were informed about the overall objective of the study, the voluntary participation, and their rights to withdraw from the study. Recruitment was subject to obtaining written informed consent. A total of 153 patients were interviewed but only 132 consented to collect samples from them. The respondents were divided into two groups of children and adults, where for children, two subgroups were under five years old and above five years up to 13. Adults were further divided into four subgroups; teenagers (13–19), youths (20–21), adults (22–60) and elders (60+).

### 2.4 Inclusion and exclusion criteria

Patients with any of the main signs/symptoms suggestive of *Salmonella* spp. Infections, including abdominal pain, diarrhoea and blood in stool, were enrolled for sample collection. Patients who contracted gastroenteritis of any cause after admission and patients who did not have gastrointestinal alterations were excluded from the study.

### 2.5 Stool sample collection

Stool samples were collected from outpatients and hospitalised patients attending MRRH during the study period. The stool samples were collected from patients admitted in the adult male recovery ward, adult female recovery ward, infectious pediatric ward, general pediatric ward, maternity ward and the laboratory. Screw-caped clean plastic containers were used for the collection of stool samples. The samples were transported to Microbiology Laboratory at the College of Veterinary Medicine and Biomedical Sciences, SUA, in an icebox within 3–4 hours of collection. They were processed upon arrival within a maximum of two hours.

### 2.6 Conventional identification of *Salmonella* spp

In the laboratory, stools were enriched in peptone-buffered water and inoculated in Selenite Fecal Broth (SFB) (Becton-Dickinson-USA) overnight. Briefly, 1 g of stool sample was enriched in 9 ml of buffered peptone water (OXOID, UK) and inoculated in Selenite Fecal Broth (SFB) (Becton-Dickinson-USA), and incubated overnight for 24 h at 37˚C. The enriched samples were cultured in Salmonella-Shigella Agar (SSA) and Xylose-Lysine-Deoxycholate (XLD) agar, according to ISO 6579 and ISO 21567 [17]. All suspected *Salmonella* spp. colonies

were picked and then subcultured on MacConkey and blood agar and incubated at 37°C for 18–24 hours. The bacteria grown were studied for their micro-morphological characteristics, while Gram stains were performed to investigate their micro-morphological characteristics as well. Conventional biochemical tests which involve several sugars, such as Triple Sugar Iron (TSI) and enzymatic tests were performed to identify the *Salmonella* spp presumptively. Both slide agglutination using polyvalent antisera (Poly A-E+Vi from SSI, Denmark) and Minibact-E biochemical tests (SSI, Denmark) were used to confirm isolates. The WHO National Salmonella and Shigella Center, Institute of Health in Bangkok, Thailand, used serotyping to identify specific serovars [18]. Phage typing was done for *S.* Typhimurium (5 isolates) and *S.* Enteritidis (6 isolates) according to the scheme defined by the PHLS Colindale London at the OIE-National reference laboratory for *Salmonella*, Instituto Zooprofilattico Sperimentale delle Venezie, Italy.

## 2.7 Molecular confirmation of *Salmonella* spp

**DNA extraction.** According to Carriero et al. (2016) [19], the thermal extraction technique was used to isolate the genomic DNA. A few colonies were briefly emulsified in 1.0 mL of nuclease-free water (provided by Inqaba Biotech, Hatfield, South Africa), washed by vortexing, and re-suspended in nuclease-free water. These steps were completed by boiling in a water bath at 95°C for 5 minutes, and then the colonies were immediately placed in ice for 5 minutes. After repeating this technique, the suspension was centrifuged at 10,000 g for 10 minutes. The total genomic DNA was spectrophotometrically quantified using NanoDrop TM Lite Spectrophotometer (Thermo Scientific, Waltham, and U.S.A) and stored at -20°C until further use [19].

**PCR for the 16S rRNA gene.** The amplification of the 1500bp fragment of the 16S rRNA gene was conducted by PCR using the universal bacterial primers for 16S rRNA 9F5'-GAGTT TGATCCTGGCTCAG-3', 1542R5'-AGAAAGGAGGTGATCCAGCC-3' in 20μl reaction mixture: PCR master mix (Bioneer, South Korea) constituted of 18μl of the reaction mix and 2μl of a DNA sample [20]. The PCR protocol was set as follows: denaturation at 94°C for 30 seconds, annealing at 58°C for 30 seconds, elongation at 72°C for 2 minutes, all these temperatures undergone 35 cycles and final extension at 72°C for 10 minutes. The post-PCR products were run in the gel electrophoresis using 1% agarose gel and 1xTE buffer for 45 minutes at 100 V. The 16S rRNA post-PCR products were sequenced using Sanger sequencing technology (ABI sequencer) at Macrogen Europe, Netherlands. Nuclease-free water was used as a negative control, and DNA extracted from a reference standard *Salmonella* Typhimurium (ATCC 13311) collected from Minnesota, USA, was used as a positive control.

**2.8 Antimicrobial susceptibility testing.** Utilising Kirby-Bauer's disc diffusion method, antimicrobial susceptibility testing was carried out using the clinical and laboratory standards institute guidelines, [21]. The tops of well-defined colonies were touched with a sterile loop and then transferred to a tube of normal saline. The colony was emulsified on the inside of the tube near the side to prevent the colony's cells from clumping together. A 0.5 McFarland standard was used as the inoculum standard, which equates to roughly 106 CFU/mL. Muller Hinton agar plates (OXOID, UK) were used for the inoculation of *Salmonella* bacteria, which were then allowed to dry for 15 minutes. The six chosen antimicrobial disks were placed on the plates, and finally, the plates were incubated at 35°C for 24 hours. A sterile cotton swab was dipped into the inoculum, swirled several times, and then pressed firmly against the inside wall of the tube above the fluid level to remove additional inoculum from the swab. Disks were applied to the agar surface using an Oxoid disk dispenser and sterile forceps. Plates were incubated in the air at room temperature (34°C) for 18 hours with the plates inverted. The

**Table 1. Set of primers and their respective annealing temperature used in this study to detect antimicrobial resistance gene.**

| Antibiotic | Gene | Primer (5' -3') | Size | A/temp | Ref |
|---|---|---|---|---|---|
| cephalosporins | *bla* $_{TEM}$ | F-ATG AGT ATT CAA CAT TTC CG<br>R-CCA ATG CTT AAT CAG TGA GG | 858 | 50 | [23] |
| | *bla* $_{SHV}$ | F-ATG CGT TAT ATT CGC CTG TG<br>R-AGC GTT GCC AGT GCT CGA TC | 862 | 58 | [23] |
| | Universal *bla* $_{CTX-M}$ | F-5′-SCS ATG TGC AGY ACC AGT AA<br>R-5′-CCG CRA TAT GRT TGG TGG TG | 554 | 58 | [24]<br>[25] |
| ciprofloxacin | *acrA* | (R) TGCAGAGGTTCAGTTTTGACTGTT<br>(F) CTCTCAGGCAGCTTAGCCCTAA | 107 | 60 | [26] |
| gentamicin | *aac(3)-I* | (F) ACCTACTCCCAACATCAGCC<br>(R) ATATAGATCTCACTACGCGC | 169 | 60 | [26] |
| tetracycline | *tet(A)* | (F) GGTTCACTCGAACGACGTCA<br>(R) CTGTCCGACAAGTTGCATGA | 577 | 57 | [27] |
| doxycycline | *tet(B)* | (F) CCTCAGCTTCTCAACGCGTG<br>(R) GCACCTTGCTGATGACTCTT | 634 | 56 | [27] |
| sulfonamides | *Sul1* | (F) CGGCGTGGGCTACCTGAACG<br>(R) GCCGATCGCGTGAAGTTCCG | 433 | 69 | [28] |
| | *Sul2* | (F) GCGCTCAAGGCAGATGGCATT<br>(R) GCGTTTGATACCGGCACCCGT | 293 | 69 | |

following standard antimicrobial disks (OXOID, UK) were used: trimethoprim (15 μg), cefotaxime (15μg), ampicillin (15 μg), tetracycline (15 μg), gentamicin (15 μg) and ciprofloxacin (15μg) [21]. Using callipers, the zones of inhibition in the cultures were quantified and evaluated. Interpretation of the results were recorded as sensitive (S), intermediate (I) or resistant (R) based on the National Committee for Clinical and Laboratory Standards Institute criteria [21]. To ensure the testing performance of the effectiveness of antimicrobial discs and the caliber of the media, the standard reference strain of *Escherichia coli* (ATCC-25922) was utilised. The resistant group also comprised intermediate resistant strains. Non-susceptibility to three or more types of antimicrobials was the definition of multidrug resistance (MDR) [22].

**2.9 Amplification of antimicrobial-resistant genes.** Specific PCRs targeting 9 antimicrobial-resistant genes were performed on each pooled DNA sample using specific primers, as shown in Table 1 below. Two Duplex PCRs were conducted to detect the *Sul1* and *Sul2* genes and *bla* $_{CTX-M}$ and *bla* $_{SHV}$ genes. Singleplex PCR was performed on the rest of the antimicrobial genes. All amplifications were performed using the same regimen except for annealing temperatures, which were adjusted based on the manufacturer's recommendations (Table 1). The amplification regimen included an initial denaturation step at 95˚C for 15 minutes, 30 cycles of denaturation at 94˚C for 30 seconds, annealing at respective temperatures for 30 seconds, extension at 72˚C for 2 minutes, followed by a final extension step at 72˚C for 10 minutes. *E. coli ATCC 25922* and Universal *bla* $_{CTX-M}$ were used as negative and positive controls, respectively. The PCR amplifications were electrophoresed in 1.5% agarose gel at 110V for 60 minutes and viewed in gel Tran's illuminator machine.

## 2.10 Data analysis

Data were analysed using STATA version 16 of 2018, where descriptive and inferential statistical analyses to descpribe the association of symptoms to Salmonella infection and demographic data were performed using Chi-squire. Descriptive statistics were used to present the social demographic characteristics of respondents. The data analysis was performed separately for children under 05 years old and individuals aged >05 years old. Descriptive statistical analysis was conducted using GraphPad Prism 9 software. The sequence comparison was

performed by BLASTING the sequences in www.ncbi.nlm.nih.gov/ BLAST to obtain the reference sequences from the gene bank. The sequence assembly from this study was conducted using Bioedit software version 7.2. All sequences (from this study and references) were aligned using MEGA XI software. The neighbor-joining phylogenetic tree was constructed using the same software.

## 3. Results

### 3.1 Demographic characteristics of the participants

One hundred and fifty-three participants involved in this study comprised two broad groups of children (n = 65) and adults (n = 88). While all the participants consented to an interview, only 132 patients consented to collect stool samples of these patients with gastrointestinal problems admitted and outpatients at the hospital originating in Morogoro for analysis. Most of the patients (74 patients) came from rural areas, but some were from the Morogoro Municipality (58 patients). The sampled patients were in the following groups: children (n = 54) and adults (n = 78), further classified as teenagers (n = 6), younger (n = 4), adults (n = 59) and elders (n = 9). Of those patients, 58/132 (44%) were male and 74/132 (56%) were female. In this study, age groups were classified as follows: children (5–8 years), teenagers (9–16 years), younger (17–21 years), adults (22–60 years) and elders (above 60 years).

### 3.2 Isolation and molecular confirmation of *Salmonella* spp

Of 132 samples subjected to conventional identification, 11(8.3%) were presumptively identified as *Salmonella* spp. The bacteria were non-lactose fermenters on MacConkey agar, while their colonies appeared colourless with black centres on SSA and pink with a black centre on XLD. The microscopic findings revealed them as short rods in single, gram-negative bacteria. Molecular amplification of the 16S rDNA gene detected a PCR product band of 1500bp in gel electrophoresis (Fig 1). Sample number 11 contains multiple bands, which makes it appear different from the rest of the bands. These multiple bands are caused by low annealing temperatures and short annealing times. But sometimes, the nature of the sample itself can cause this condition to happen. Further optimization of PCR is supposed to be done to give good results. The results for phylogenetic analysis displayed two important non-typhoid *Salmonella* serotypes; *Salmonella* Enteritidis (6 isolates) and *Salmonella* Typhimurium (5 isolates) from several sequences in the gene banks (Fig 2). The sequences from this study showed a percentage identity of 97–99% during blasting with those used in phylogenetic analysis.

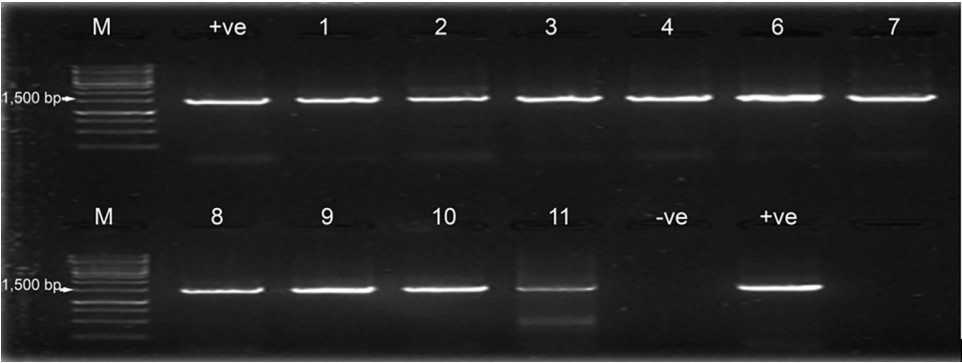

**Fig 1. PCR amplification of 16S rDNA gene (1500 bp) of the 11 isolates: M = Marker DNA size Marker (1500bp), +ve = Positive control, 1–11 = Samples, -ve = Negative control.**

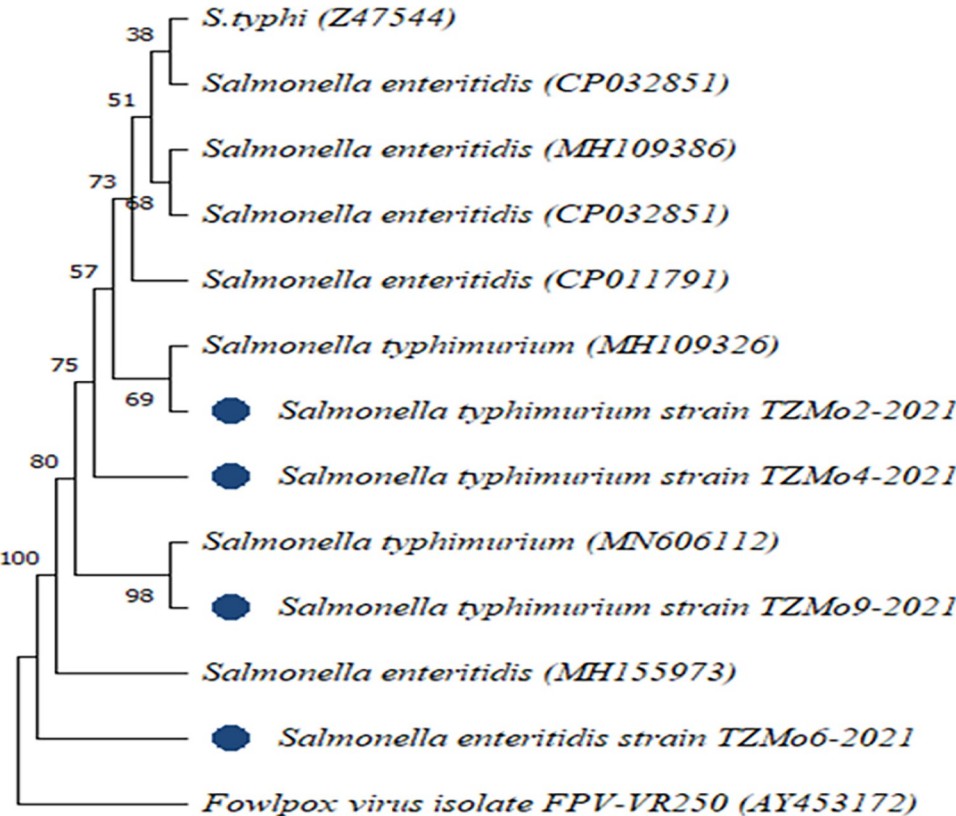

**Fig 2. Phylogenetic tree for *Salmonella* serotypes from this study (blue circle) and closely related taxa from the gene bank to assess the genetic similarity of the *Salmonella* isolates from human faeces from Morogoro.** The tree was generated using the Neighbor-Joining method (p-distance model), with bootstrap values expressed as percentages of 1,000 replications. *Fowlpox* virus (AY453172) served as an out-group.

### 3.3 The occurrence of *Salmonella* spp between children and adults

A comparison of the isolation rate between adults and children showed a higher infection rate in children 5.3% (7/132) than in adults 3.0% (4/132); however, the difference (P>0.05) was not statistically significant.

### 3.4 Phenotypic and molecular antimicrobial susceptibility assessment of isolated *Salmonella* spp

The results of antimicrobial susceptibility profiles of the *Salmonella* spp. isolates showed that all of the total isolates, 100% (n = 11), were resistant to at least one antimicrobial. It further showed that all of the *Salmonella* spp. isolates were resistant to ampicilin (100%, n = 11). The other antimicrobial resistance occurrences were cefotaxime (36.4%, n = 4), tetracycline and tri-methoprim (27.3%, n = 3 each). Resistance to gentamicin and ciprofloxacin was 9.1% each, as observed in Table 2. The results of antimicrobial susceptibility profiles of the *Salmonella* spp. isolates showed resistance to one or more classes of antimicrobials, taking into account the Salmonella isolates tested as multiple drug resistance isolates (MDR), as observed in Table 3. There was a marked difference in resistance between the two antimicrobials (gentamicin and ciprofloxacin) and the rest of the tested antimicrobials (*p<-0.05)*. Nine different resistant genes were detected using specific primers, as shown in Table 1. The genes include the cepha-losporin-resistant gene (*bla*$_{TEM}$ gene, *bla*$_{SHV}$ gene and universal *CTX*-$_M$ gene), ciprofloxacin-

**Table 2. Antimicrobial resistance profiles of the *Salmonella* isolates.**

| ANTIMICROBIALS | | FREQUENCY | PERCENTAGE (%) |
|---|---|---|---|
| **Tetracycline** | I | 1 | 9.1 |
| | R | 3 | 27.3 |
| | S | 7 | 63.6 |
| **Ampicilin** | R | 11 | 100.0 |
| **Cefotaxime** | I | 1 | 9.1 |
| | R | 4 | 36.4 |
| | S | 6 | 54.5 |
| **Gentamicin** | I | 3 | 27.3 |
| | R | 1 | 9.1 |
| | S | 7 | 63.6 |
| **Trimethoprim** | R | 3 | 27.3 |
| | S | 8 | 72.7 |
| **Ciprofloxacin** | I | 6 | 54.5 |
| | R | 1 | 9.1 |
| | S | 4 | 36.4 |

Key: I = Intermediate resistant, R = Resistant and S = Susceptible

resistant gene (*arc* gene), gentamicin-resistant gene (*arc(3)-I* gene), tetracycline-resistant gene (*tetA* gene), doxycycline resistant gene (*tetB* gene) and sulfamethoxazole-trimethoprim resistant gene (*Sul 1* and *Sul 2* gene). Sulfonamide-resistant genes were highly detected (*Sal1* = 100%) compared with ciprofloxacin (0) and gentamicin (0) Table 3. The agarose gel pictures showing the corresponding band size have not been shown.

## 3.5 Association between *Salmonella* symptoms and patient age category

The symptoms diagnosed included diarrhoea, blood in stool, nausea, vomiting, fever, and general pains. Results showed that nausea, vomiting and general aches and pains were significantly associated with age category (p<0.01). Other symptoms like diarrhoea, blood in stool and fever had no significant association with age of the patient (Table 4).

**Table 3. Phenotypic and genotypic resistance proportions displayed by the identified *Salmonella* spp.**

| Antimicrobial category | Antimicrobial agent | Percentage phenotypic resistance (N = 11) | Percentage Resistance genes |
|---|---|---|---|
| Aminoglycosides | gentamicin | 9.1% (n = 1) | *Acc(3)-I* (n = 0, 0.0%) |
| Extended-spectrum cephalosporins; 3rd and 4th generation cephalosporins | cefotaxime | 36.4% (n = 4) | $Bla_{CTXM}$ (n = 18, 18%) $Bla_{SHV}$ (n = 2, 18%) $Bla_{TEM}$ (n = 3, 27%) |
| Fluoroquinolones | ciprofloxacin | 9.1% (n = 1) | *acrA* (n = 0, 0.0%) |
| Folate pathway inhibitors | trimethoprim-sulphamethoxazole | 27.3% (n = 3) | *Sal1* (n = 11, 100%) *Sal2* (n = 6, 54%) |
| Penicillins | ampicillin | 100% (n = 11) | $Bla_{CTXM}$ (n = 18, 18%) $Bla_{SHV}$ (n = 2, 18%) $Bla_{TEM}$ (n = 3, 27%) |
| Tetracyclines | tetracycline | 27.3% (n = 3) | *Tet A* (n = 2, 18%) *Tet B* (n = 0, 0.0%) |
| More than two antimicrobial category | MDR | 27.3% (n = 3) | - |

**Table 4. Association between patient's age category and *Salmonella* symptoms (n = 153).**

| Salmonella Symptoms | | Patient category | | | | Chi-square value ($X^2$) | P-value |
|---|---|---|---|---|---|---|---|
| | | Adult | | Children | | | |
| | | Count | Percentage | Count | Percentage | | |
| **Diarrhoea** | +VE | 26 | 29.5 | 29 | 44.6 | 3.687 | 0.055 |
| | -VE | 62 | 70.5 | 36 | 55.4 | | |
| | | 88 | 100 | 65 | 100 | | |
| **Blood in stool** | +VE | 2 | 2.3 | 3 | 4.6 | 0.649 | 0.420 |
| | -VE | 86 | 97.7 | 62 | 95.4 | | |
| | | 88 | 100 | 65 | 100 | | |
| **Nausea (feeling sick)** | +VE | 34 | 38.6 | 6 | 9.2 | 17.685 | 0.000*** |
| | -VE | 54 | 61.4 | 58 | 89.2 | | |
| | | 88 | 100 | 64 | 100 | | |
| **Vomiting (being sick)** | +VE | 14 | 15.9 | 23 | 35.4 | 9.414 | 0.009*** |
| | -VE | 74 | 84.1 | 41 | 63.1 | | |
| | | 88 | 100 | 64 | 100 | | |
| **Feeling feverish** | +VE | 67 | 76.1 | 41 | 63.1 | 3.071 | 0.800 |
| | -VE | 21 | 23.9 | 24 | 36.9 | | |
| | | 88 | 100 | 65 | 100 | | |
| **General aches and pains** | +VE | 62 | 70.5 | 17 | 26.2 | 34.764 | 0.000*** |
| | -VE | 23 | 26.1 | 48 | 73.8 | | |
| | | 85 | 100 | 65 | 100 | | |
| **Other symptoms** | +VE | 12 | 13.6 | 9 | 13.8 | 0.582 | 0.748 |
| | -VE | 71 | 80.7 | 54 | 83.1 | | |
| | | 83 | 100 | 63 | 100 | | |

+VE = Positive -VE = Negative * = 95% level of significance (P<0.1) *** = 95% level of significance (P<0.01)

## 4. Discussion

Non-typhoid gastroenteritis caused by *Salmonella* spp. continues to be an important public health issue that often requires hospitalisation in Morogoro. This is one of the few studies exploring the genetic diversity of salmonellosis in Morogoro for elders, adults, youth, and young children. *Salmonella* spp. is developing resistance to many used antimicrobials, which makes it relevant to this study.

The overall prevalence of *Salmonella* spp in Morogoro region was 8.3% (11/132) (95% C.I; 5%-10.0%), with *Salmonella* Enteritidis *and Salmonella* Typhimurium being the only serovars detected from adults and children stool. A comparison of the isolation rate showed that children had a higher proportion (11.6%) compared to adults (10.5%). However, the difference was not statistically significant (P>0.05). This is concomitant to studies from Mozambique, where a high burden of NTS was reported among young children (*S*. Typhimurium and *S*. Enteritidis) [29]. In most parts of the world, surveys have reported *S*. Enteritidis and *S*. Typhimurium as the major serovars found in humans [30,31]. Feasey et al. [32] also reported 32% compared with 54% of invasive NTS in children <15 years of age in South Africa and Malawi, respectively. In Côte d'Ivoire, the mortality rate was estimated to reach 5% from *Salmonella* infections. From studies carried out between 2005 and 2009 among isolated serotypes, NTS was prevalent [33]. It was noted from this study that *S*. Enteritidis is most dominant among hospital isolates, while *S*. Typhimurium serovar was the least dominant. The previous studies conducted in Mukuru slum, an urban informal settlement in Kenya, reported *S*. Typhi was the

dominant serovar followed by *S*. Typhimurium and *S*. Enteritidis [34]. Previous studies in other sub-Saharan African countries also showed the dominance of *S*. Typhimurium [35]. Different serovars of *Salmonella* spp. are known to be commonly detected in a given area at different times depending on the type of serovars of *Salmonella* circulating in food sources in a study area [36]. In most other African countries, these serovars are the most frequently isolated from diarrheal diseases, which is confirmed by the present study's findings. In Tanzania there are scacy data published on contribution of non typhoidal *Salmonella* spp in causing gastroentelitis and their antimicrobial status in human, howover, a considerable literature is available in food animals and food products.

NTS resistance to the commonly used antimicrobials calls for the implementation of national surveillance systems for antimicrobial resistance and implementation of prudent antimicrobial usage, including revision of current national guidelines for antimicrobial use. *S*. Typhimurium has often been associated with multiple antimicrobial resistances [37], partly due to the emergence of *S*. Typhimurium definitive phage type (DT) 104 worldwide. Strains of this phage type are resistant to ampicillin, chloramphenicol, streptomycin, sulfonamides, and tetracycline [38]. Phage typing has proven helpful for classifying *S*. Enteritidis and *S*. Typhimurium strains from various sources. However, the technique has shown to be inadequately selective when just one phage type has predominated, necessitating further characterisation.

The findings in this study reported a low cephalosporin resistance but high resistance to other antimicrobials among *Salmonella* strains. It was indicated that the resistance was higher to ampicillin (100%) than all other antimicrobials. The antimicrobial resistance in *Salmonella* spp. is one of the main concerns when it comes to the issue of human infection treatment [39]. Moreover, the high resistance of *Salmonella* spp. to ampicillin was also recorded by a study from Harare, where *Salmonella* spp. was resistant to ampicillin (100%), as reported by Dinkineh et al. [40]. A similar finding was reported by Bilal et al. [41] in Pakistan. It is also important to note that, a study from Addis Ababa reported that 81.2%, 94.5%, and 75.7% of the isolates were resistant to ampicillin, tetracycline, and sulfamethoxazole-trimethoprim, respectively [42].

In contrast to the present study, 72.7%, 63.6% and 54.5% of *Salmonella* spp. isolates were shown to be susceptible to sulfamethoxazole-trimethoprim, tetracycline and cefotaxime respectively. It is also worth noting that, generally the recovered *Salmonella* spp. serovars in this study showed low resistance to cephalosporins and high resistance to other antimicrobials routinely used for the treatment of humans in Tanzania. The increased resistance towards these commonly used antimicrobials may have been contributed by misuse, incorrect diagnosis and failure to abide by treatment guidelines. This investigation identified a significant trend in the emergence of MDR in *Salmonella* Typhimurium and *Salmonella* Enteritidis isolates. The spectrum of resistance in NTS isolates used in this study extended further to include cefotaxime, trimethoprim-sulphamethoxazole and tetracycline, with the resistance levels agreeing with rates found in a study carried out in Thailand by Toni et al., (2018) [43]. In this study, the antimicrobial resistance gene results found that 8 isolates in adults and children contain sulfamethoxazole (*Sul 1* and *2*) resistance gene, tetracycline (*tetA*) resistance gene, $Bla_{CTXM}$ resistance gene, $Bla_{SHV}$ resistance gene and $Bla_{TEM}$ resistance gene. In contrast, no gene was detected for *tet(B)*, *Acc(I-3)* and *AcrA*. These results agree with those of Hedayatianfardi et al., [44]. who found no genetic evidence for tet(B) presence in their experiments using PCR. However, phenotypic results showed resistance to tetracycline and sulfamethoxazole, but only sulfamethoxazole carried the resistant gene. This is because the phenotype of most isolates is influenced by specific and non-specific resistance mechanisms such as lower membrane permeability and high active efflux pumps [45]. In the current study, the isolates carrying the *tetA* gene were phenotypically susceptible to tetracycline. Tetracycline resistance in bacteria is

mediated by a group of *tet* genes through different mechanisms, including ribosomal protection and efflux pumps, [46]. On the other hand, *tet* genes have different modes of action, which, in some cases, necessitate the presence of more than one type of *tet* gene for the bacteria to fully express phenotypic antimicrobial resistance [47]. The lack of consistency in the relationship between the presence of resistance genes and phenotypic antimicrobial resistance of the isolates observed in the current study indicates the importance of accompanying the resistance gene analysis with phenotypic antimicrobial resistance testing.

Based on *Salmonella* spp. symptoms and patient category, adults and children suffering from *Salmonella* had more episodes of fever and few of them complained about visible blood in the stool. The difference, however, was statistically insignificant. The symptoms like diarrhoea, blood in stool and fever had no significant association with the age of the patient in Morogoro region. Such observations are concomitant with the findings of other studies [48] Diarrhoea was significantly related to NTS infection in the Morogoro region. However, it is important to note that these clinical symptoms might be caused by other factors, including changes in diet, which can even affect faecal consistency.

## 5. Conclusion

In comparison to clinical and microbiological characteristics of patients with gastrointestinal problems at the hospital in Morogoro, the infection rate of 8.3% NTS was established. These results, combined with a different antimicrobial susceptibility profile, warrant further investigation into the epidemiology of NTS infections. NTS is one of the more important causes of illness in children than adults in Morogoro region, especially among children less than 5 years of age, implicated to their immunity. To target the control actions to the critical points in the transmission of the pathogens to consumers, systematic surveillance of the infection sources in humans and animals, as well as the routes of the bacterial pathogens, is therefore mandatory.

## 6. Study limitation

This study had some limitations. Firstly, it was limited by information bias. Questions for children were designed to enable parents or guardians to report on their behalf (recall bias), thus underestimating the true ORs or association of causal effect. Another limitation of this study was the use of stool samples only; multiple sampling sources, such as blood, could have created a nice ground for the broad prevalence establishment.

## Acknowledgments

This work would not have been accomplished without the assistance of SUA, MRRH and Doctors staff, and Laboratory scientists of Department of Microbiology Laboratory at the College of Veterinary Medicine and Biochemical Sciences, SUA, for analyses of samples. I appreciate the parents or guardians' contribution to this study. Additionally, I would like to thank the Morogoro Regional Referral Hospital's medical personnel, nursing staff, and administrative offices for recruiting participants. Additionally, I recognise the efforts of individuals who contributed to data collection with compensation. I am also grateful to Prof. Gerald Misinzo, who assisted in ethics application and subject recruitment for the National Institute of Medical Research, and Dr. George Alcard Rweyemamu, who assisted in Clinical Management System checking at Morogoro Regional Referral Hospital.

## Author Contributions

**Conceptualization:** Anacleta Cuco, Robinson Mdegela.

**Data curation:** Anacleta Cuco, Alexanda Mzula.

**Formal analysis:** Anacleta Cuco, Alexanda Mzula.

**Methodology:** Anacleta Cuco.

**Supervision:** Ernatus Mkupasi, Alexanda Mzula, Robinson Mdegela.

**Validation:** Ernatus Mkupasi.

**Visualization:** Ernatus Mkupasi.

**Writing – original draft:** Anacleta Cuco.

**Writing – review & editing:** Alexanda Mzula, Robinson Mdegela.

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
