## [Decision Letter · Decision Letter 0]

8 May 2024

Dear Dr. Mzula,

Thank you very much for submitting your manuscript "Non-Typhoid Salmonella Contributes to Gastrointestinal infections in Morogoro; evidence from Patients Attending Morogoro Regional Referral Hospital in Tanzania" for consideration at PLOS Neglected Tropical Diseases. As with all papers reviewed by the journal, your manuscript was reviewed by members of the editorial board and by several independent reviewers. The reviewers appreciated the attention to an important topic. Based on the reviews, we are likely to accept this manuscript for publication, providing that you modify the manuscript according to the review recommendations. 

Sincerely,

Abiola Senok, MBBS; PhD, FRCPath

Academic Editor

Mathieu Picardeau

Section Editor

Reviewer's Responses to Questions

**Key Review Criteria Required for Acceptance?**

**Methods**

-Are the objectives of the study clearly articulated with a clear testable hypothesis stated?

-Is the study design appropriate to address the stated objectives?

-Is the population clearly described and appropriate for the hypothesis being tested?

-Is the sample size sufficient to ensure adequate power to address the hypothesis being tested?

-Were correct statistical analysis used to support conclusions?

-Are there concerns about ethical or regulatory requirements being met?

Reviewer #1: I think the objective and hypothesis of the study were well presented and addressed.

The sample size was adequate basing on the fact that they only worked only on one station (Morogoro referral Hospital). The statistical analysis was adequate

Reviewer #2: Yes for the following questions:

-Are the objectives of the study clearly articulated with a clear testable hypothesis stated?

-Is the study design appropriate to address the stated objectives?

-Is the population clearly described and appropriate for the hypothesis being tested?

-Is the sample size sufficient to ensure adequate power to address the hypothesis being tested?

-Were correct statistical analysis used to support conclusions?

-Are there concerns about ethical or regulatory requirements being met?: No.

**Results**

-Does the analysis presented match the analysis plan?

-Are the results clearly and completely presented?

-Are the figures (Tables, Images) of sufficient quality for clarity?

Reviewer #1: This part of the manuscript is well presented (data analysis to generate tables and other figures was adequate)

Reviewer #2: Yes for the following questions.

-Does the analysis presented match the analysis plan?

-Are the results clearly and completely presented?

-Are the figures (Tables, Images) of sufficient quality for clarity?

**Conclusions**

-Are the conclusions supported by the data presented?

-Are the limitations of analysis clearly described?

-Do the authors discuss how these data can be helpful to advance our understanding of the topic under study?

-Is public health relevance addressed?

Reviewer #1: Yes the authors presented their ideas on the conclusion, however I think they need to be specific especially they mention of the stewardship campaigns without stating specifically where could the stewardship be conducted

Reviewer #2: -Are the conclusions supported by the data presented?

-Are the limitations of analysis clearly described?

-Do the authors discuss how these data can be helpful to advance our understanding of the topic under study?

-Is public health relevance addressed?

**Editorial and Data Presentation Modifications?**

Reviewer #1: Minor revision, some comments have been embedded in the man document

Reviewer #2: Minor corrections: 

1. Line 23: Salmonella spp.............(italicise Salmonella). 

2. Line 39: Correct spelling of serovars.............

3. Intro: Cite references such as - https://doi.org/10.1099/jmm.0.022319-0. PMID: 20813852

4. Either use the term serovar or serotype, not both. 

5. Line 117:" Change the word stomached. 

6. Line 135: error in spelling- consented?

7. Line 176: Should be subcultured ON......

8. References 20-22, for AST and CLSI cite more relevant and latest references. 

9. For CLSI reference: Please clarify the year of reference, mainly for AST interpretations and MIC breakpoints. 

10. Line 275: Correct the spelling of- broader...

11. Discussion: Include global, regional and then local data for comparison. 

12. In discussion: Names of strains needs to follow uniform way - S. Enteritidis etc. 

13. The manuscript needs to be corrected for its language by a native english speaker.

**Summary and General Comments**

Reviewer #1: Some comments have been included in the pdf version of the manuscript

Reviewer #2: Please cite and list most relevant and latest references. 

Please correct the errors stated. 

The manuscript can be accepted, provided all the stated corrections are done. 

The manuscript needs to be corrected for its language by a native english speaker.

PLOS authors have the option to publish the peer review history of their article (what does this mean?). If published, this will include your full peer review and any attached files.

Reviewer #1: Yes: Philbert Balichene Madoshi

Reviewer #2: Yes: Dr. Godfred Antony Menezes

Figure Files:

Data Requirements:

Reproducibility:

References

---

## [Editor Report · Decision Letter 1]

27 May 2024

Dear Dr. Mzula,

We are pleased to inform you that your manuscript 'Non-Typhoid Salmonella Contributes to Gastrointestinal infections in Morogoro; evidence from Patients Attending Morogoro Regional Referral Hospital in Tanzania' has been provisionally accepted for publication in PLOS Neglected Tropical Diseases.

Best regards,

Abiola Senok, MBBS; PhD, FRCPath

Academic Editor

Mathieu Picardeau

Section Editor

---

## [Editor Report · Acceptance letter]

31 May 2024

Dear Dr. Mzula,

We are delighted to inform you that your manuscript, "Non-Typhoid Salmonella Contributes to Gastrointestinal infections in Morogoro; evidence from Patients Attending Morogoro Regional Referral Hospital in Tanzania," has been formally accepted for publication in PLOS Neglected Tropical Diseases.

Best regards,

Shaden Kamhawi

co-Editor-in-Chief

Paul Brindley

co-Editor-in-Chief
